# A Review of the Repair of DNA Double Strand Breaks in the Development of Oral Cancer

**DOI:** 10.3390/ijms25074092

**Published:** 2024-04-07

**Authors:** Stephen S. Prime, Piotr Darski, Keith D. Hunter, Nicola Cirillo, E. Kenneth Parkinson

**Affiliations:** 1Centre for Immunology and Regenerative Medicine, Institute of Dentistry, Barts and the London School of Medicine and Dentistry, Queen Mary University of London, London E1 4NS, UK; stephensprime@gmail.com; 2Liverpool Head and Neck Centre, Molecular and Clinical Cancer Medicine, University of Liverpool, Liverpool L69 3BX, UK; p.darski@liverpool.ac.uk (P.D.); keith.hunter@liverpool.ac.uk (K.D.H.); 3Melbourne Dental School, University of Melbourne, 720 Swanson Street, Carlton, Melbourne, VIC 3053, Australia; nicola.cirillo@unimelb.edu.au; 4School of Dentistry, University of Jordan, Amman 11942, Jordan

**Keywords:** oral cancer development, DNA repair, double strand breaks, homologous recombination, Fanconi anemia, non-homologous end joining

## Abstract

We explore the possibility that defects in genes associated with the response and repair of DNA double strand breaks predispose oral potentially malignant disorders (OPMD) to undergo malignant transformation to oral squamous cell carcinoma (OSCC). Defects in the homologous recombination/Fanconi anemia (HR/FA), but not in the non-homologous end joining, causes the DNA repair pathway to appear to be consistent with features of familial conditions that are predisposed to OSCC (FA, Bloom’s syndrome, Ataxia Telangiectasia); this is true for OSCC that occurs in young patients, sometimes with little/no exposure to classical risk factors. Even in Dyskeratosis Congenita, a disorder of the telomerase complex that is also predisposed to OSCC, attempts at maintaining telomere length involve a pathway with shared HR genes. Defects in the HR/FA pathway therefore appear to be pivotal in conditions that are predisposed to OSCC. There is also some evidence that abnormalities in the HR/FA pathway are associated with malignant transformation of sporadic cases OPMD and OSCC. We provide data showing overexpression of HR/FA genes in a cell-cycle-dependent manner in a series of OPMD-derived immortal keratinocyte cell lines compared to their mortal counterparts. The observations in this study argue strongly for an important role of the HA/FA DNA repair pathway in the development of OSCC.

## 1. Introduction

Oral squamous cell carcinoma (OSCC) is the most common malignancy of head and neck squamous cell carcinomas (HNSCC), and globally, it accounts for some 377,000 new cases and 177,000 deaths per annum [1]. OSCC develops in older male adults (>65 years) who invariably are exposed to tobacco, alcohol, and/or areca nut [2]; oropharyngeal cancer is also associated with human papilloma virus (HPV) infection [3]. Unfortunately, OSCC has also been reported in young individuals (<50 years of age; [4]) who do not present with longstanding histories of exposure to classical risk factors [5], despite a global decline in the use of tobacco and alcohol in the developed world [6]. The cause of OSCC therefore remains an enigma. 

Oral potentially malignant disorders (OPMD) can precede the development of OSCC and manifest as white (leukoplakia) and/or red (erythroplasia/erythroplakia) lesions of the oral mucosa that cannot be attributable to any other recognizable condition [7,8]. OPMD share many of the epidemiological characteristics with OSCC—they occur predominantly in males, are associated with tobacco use, and are a feature of elderly individuals in developed countries (40–80 years); they occur 5–10 years earlier in individuals from developing countries. Overall, the transformation rate for OPMD (all categories) is 7.9%, but significant predictors include non-smoking status, site and size of lesion, a non-homogenous appearance, and the degree of epithelial dysplasia [9]. Variations in these parameters are likely to reflect the nature of the participating individuals (age, gender, co-morbidities), the extent of exposure to risk factors, and the duration of follow-up in specific research studies.

A variety of OPMDs other than leukoplakia/erythroplasia also have a propensity for malignant changes, including oral submucous fibrosis, oral lichen planus, lichenoid lesions, palatal lesions in reverse smokers, chronic graft versus host disease, and lupus erythematosus [10]. Many of these disorders are associated with disease-specific pathological mechanisms, and unless relevant, they will not be considered further in this review.

Cancer development has always been described as a continuum of sequential stages from normality to malignancy. Clinicians, for example, describe this transition as normal mucosa to pre-malignancy to invasive carcinoma, pathologists invoke the terms mild/moderate/severe dysplasia to invasive carcinoma, and cell/molecular biologists view the transition as the progressive accumulation of multiple gene anomalies that lead to the selection of more dominant cell phenotypes. We have discussed the limitations of this step-wise approach and argued strongly that genomic instability plays a key role in the development of oral malignancy [11,12]. 

The primary aim of this review is to investigate whether defects in the response and repair of DNA double strand breaks (DSBs) predispose OPMD to undergoing a malignant change to OSCC. To address this question, we examined whether (1) defects in the recognition and repair of DSBs occur commonly in OPMD; (2) familial conditions with defects of DNA repair pathways predispose individuals to develop OSCC; (3) abnormalities of DSB repair genes explain the early onset of OSCC in young patients with limited exposure to classical OSCC risk factors; and (4) anomalies in the expression of DSB repair genes are present in OPMD-derived cultured keratinocytes with known clinical outcomes.

## 2. DNA Damage

DNA damage occurs due to exogenous and/or endogenous factors. Exogenous causes include exposure to environmental, physical, and chemical factors such as UV and ionizing radiation, together with alkylating and cross-linking agents. By contrast, endogenous agents include replication stress, inadvertent cleavage by nuclear enzymes, hydrolysis and oxidation of chemically active DNA, and naturally occurring reactive oxygen species.

DNA damage can lead to genomic instability, which describes a spectrum of genetic alterations ranging from small nucleotide changes (mutations, insertions, deletions) to extreme chromosomal alterations. In the present review, we focus on chromosome instability (CIN) that can be defined as an increase in the rate of chromosomal change manifesting as both numerical and structural alterations. Numerical CIN is associated with gains and losses of whole chromosomes due to mis-segregation of chromosomes during mitosis, whereas structural CIN is characterized by amplifications, deletions, inversions, duplications, and balanced/unbalanced translocations. CIN is generated by the incomplete/deficient repair of DSBs, by critically shortened telomeres that are recognized as DSBs, and by defects in cell cycle checkpoint genes. DSBs are highly toxic and, arguably, they present the greatest challenge to cell viability. Approximately 10–50 DSBs occur in any given cell per day, depending on the specific tissue [13]. 

Other terms that are commonly used in this text include aneuploidy (state of chromosome number rather than the rate of change as seen in structural CIN), and somatic chromosome number alterations (SCNA).

## 3. Causes of DSBs

The pathological formation of DSBs is commonly related to ionizing radiation, chemotherapeutic drugs (alkylating agents, cross-linking agents, radiomimetics), DNA replication stress, and defects in transcription. With regard to the aetiology of oral cancer, tobacco smoking [14], areca nut use [15], alcohol intake [16], microbial infection [17,18], chronic inflammation [19], and the production of reactive oxygen species (ROS; [20]) have all been shown to induce DSBs. However, the molecular profile of oral cancer is varied. Lung cancer is a tobacco-related cancer and is associated with guanine to thiamine (G > T) transversions, whereas there is no enrichment of G > T transversions in head and neck cancer [21]. Carcinogens such as alkylating agents cause guanine to adenine (G > A) transitions. There is overlap in the molecular profile of lung, head, and neck cancer; however, because oxidative damage is common to both cancers, if incorrectly repaired, it leads to G > T or G > C transversions as well as larger deletions [22]. Taken together, the data suggest that the causative carcinogens in the oral cavity are diverse in nature and not just the product of tobacco use. 

## 4. Consequences of DSBs

(1)Ageing

There is a plethora of information linking DSBs with the ageing process. The evidence is based on the expression of surrogate markers of DSBs in ageing human and animal cells and tissues, as well as decreases in the repair of DSBs during the ageing process; there are studies relating to inherited premature ageing syndromes and the expression of telomere-associated DDR foci and telomere-induced DDR foci during interventions known to increase health life span, or during physiological states known to accelerate ageing [23,24]. 

(2)Programmed cell death

When the rate of DNA damage exceeds the repair capacity of a cell (like homologous recombination and non-homologous end joining), the removal of irreversibly damaged cells occurs efficiently by apoptosis [25]. This is a protective mechanism to prevent the propagation of damaged DNA and ensures that cells with potentially oncogenic mutations are eliminated.

(3)Cellular senescence

Cellular senescence is defined as an irreversible cell cycle arrest that is distinct from quiescence, terminal differentiation, and apoptosis; more recently, the definition has been broadened to include other forms of senescence. Senescence occurs following ageing and multiple rounds of cell division (replicative senescence). A broad spectrum of other stresses that lead to senescence have also been described, including DNA damage, oxidative damage, hypoxia, signalling imbalances, activation of oncogenes, and cancer-related therapy [26]. In the skin, the primary cell type that expresses senescent markers are ageing melanocytes, but the senescence spreads to neighboring keratinocytes by paracrine mechanisms [27], which in turn, results in a decrease in tissue proliferation [28,29]. 

In cancer, senescence is a double-edged sword. In the early stages of carcinogenesis, senescence acts as a tumor suppressor because—with activation of the adaptive immune system—senescent putative cancer cells are eliminated [30]. Later, senescent epithelial- and stromal-cells produce a myriad of pro-inflammatory and pro-survival cytokines termed the senescent associated secretory phenotype (SASP), which is pivotal in cancer development and progression [30,31,32,33,34,35,36,37]. 

(4)Cancer

The majority of cancers develop due to genomic instability that acts in association with clonal evolution and the inactivation of tumor suppressor pathways—notably, the p53 pathway [38]. In the early stages of cancer development, the accumulation of DSBs is due to replication stress, but later, current thinking is that defects in DSB repair pathways potentiate genomic instability [39,40]. Indeed, defects in DSB repair pathways have been reported in a broad spectrum of epithelial cancers [41,42,43], including OSCC [42,43,44,45,46,47].

## 5. DNA Damage Response

The DNA damage response (DDR) comprises multiple DNA repair pathways that maintain genomic integrity. The DDR works in close association with DNA repair machinery, telomere maintenance, DNA replication, and chromosome segregation. 

### 5.1. Response

Progression through various phases of the cell cycle is dependent on the activation of cyclin-dependent kinases (Cdk) by regulatory proteins termed Cyclins. Cyclin D-Cdk4/6 and Cyclin E-Cdk2 function in the G1 phase of the cell cycle, Cyclin A-Cdk2 in the S phase, and Cyclin B-Cdk1 in the G2/M phase; metaphase to anaphase transitions during mitosis are determined by the spindle checkpoint. 

Sensor proteins such as the MRN complex (*MRE11*/*RAD50*/*NBS1*) and the Ku70/Ku80 heterodimer initially respond to DNA damage and then activate ATM (ataxia telangiectasia mutated) and ATR (ATM and Rad3-related) in response to DSBs and single strand breaks, respectively. ATM/ATR phosphorylate CHK1 (via ATR) and CHK2 (via ATM), both of which then inhibit Cdc25. Cdc25 inhibition leads to the failure to activate Cdk1, and ATM/ATR also activate p53 that inhibits Cyclin B1 expression through p21^WAF1/Cip1^. The inhibition of Cyclin B1 expression and the failure to activate Cdk1 block G2/M transition. The situation is compounded first by wee1 and Myt1 expression—which block Cdk1 activation—and then by the p53-dependent expression of 14-3-3, which chelates Cdc 25, leading to its nuclear export. 

p21^WAF1/Cip1^ is a master regulator across the whole of the cell cycle. Other than the regulation of Cyclin B1, p21^WAF1/Cip1^ inhibits Cyclin A and Cyclin E, which together with their kinases, results in G1 arrest and prevention of the G1/S transition. p21^WAF1/Cip1^ also inhibits Cyclin E–Cdk2 through the prevention of pRB phosphorylation and the inhibition of E2F gene transcription. 

There are other key mediators of the DDR: (1) DNA-dependent protein kinases (DNA-PK) co-operate with ATM and ATR to phosphorylate proteins involved in cell cycle checkpoints, and later, are involved in DNA repair; (2) PARPs (poly [ADP-ribose] polymerases) catalyze the transfer of ADP-ribose to many downstream targets, including histones—ultimately, the chromatin structure is relaxed, thereby facilitating the access of DNA repair factors to the sites of DNA damage; (3) γ-H2A.X is the phosphorylated form of histone H2A genes and is an early response to DSBs; (4) p53 binding protein 1 (*53BP1*) is involved in the selection of a specific repair pathway, and favors non homologous end joining (NHEJ) over homologous recombination (HR).

The DNA damage response is commonly activated in early neoplastic lesions and is thought to protect against malignancy [48,49]. However, following breaches in the DDR barrier due to defects of DDR-related genes (somatic or germ-line mutations, single nucleotide polymorphisms, epigenetic alterations) and inactivation of the p53 pathway, there is progressive accumulation of DSBs that ultimately leads to cancer development. DDR defects have been reported in a broad spectrum of human cancers [50,51], including HNSCC [52].

### 5.2. Repair

The repair of DNA damage occurs by a variety of pathways depending on the type and severity of the damage, the cell type (mitotically active, normal somatic cell, stem cell), the stage of the cell cycle, and the chromatin status. The primary mechanisms of DSB repair are HR and NHEJ, the genes of which are listed in Table 1. HR genes act in association with Fanconi anaemia (FA) genes. A third mechanism of DSB repair has been identified, termed micro-homology-mediated end joining or alt-non-homologous end joining (A-EJ), which uses micro-homology sequences that are distant from the DSB site; it is considered to be a back-up for HR and NHEJ when the HR pathway is inactive, or when key proteins of classical NHEJ are absent. 

HR repairs DSBs by copying intact homologous DNA sequences that are used as templates to promote error-free DNA repair in the S phase of the cell cycle. Nevertheless, HR can still give rise to genome rearrangements, particularly when recombination involves homologous sequences on heterologous chromosomes or repeat elements. Whilst HR is involved in the repair of DSBs, it is essential for genome duplication, and is thus obligatory for cellular viability. To explain the apparent paradox that cells remain vital despite the presence of germline mutations in HR genes (*BRCA1*/*2* mutations associated with familial breast and ovarian cancer), cancer cells are thought to adopt strategies to maintain some level of HR activity [53]. NHEJ occurs throughout the cell cycle. DSB ends are first captured by proteins that form a scaffold for the recruitment of additional molecules; a bridge is formed that connects the DSB ends together, and there follows re-ligation of the DNA strands. NHEJ is extremely efficient but error-prone, leading to small base pair deletions and translocations.

## 6. Response and Repair of DSBs in OPMD

### 6.1. Response

Studies that have examined the expression of DDR proteins during oral carcinogenesis have produced inconsistent findings. Some reports show that there is an increase in the expression of *ATM*, *ϒ-H2A.X*, and *53BP1* during the development of OSCC [54,55], and that the expression of *ATM* and *ϒ-H2A.X* correlates with malignant transformations [56]. By contrast, other studies have reported a progressive increase in ϒ-H2A.X up until dysplasia but not thereafter [57]. We recently examined DDR proteins in mortal and immortal keratinocytes from OPMD [58] and showed that p53 phosphorylation was higher in mortal keratinocytes relative to their immortal counterparts, whereas the opposite was true for ATM phosphorylation, which was higher in immortal compared to mortal cells.

The p53 and pRB/p16^IN4A^ pathways are fundamental to cell cycle regulation and the DDR. Dysfunction of p53 (gene mutation, LOH, increased expression of MDM2, the effects of HPV E6/E7 viral oncoproteins) and pRB/p16^IN4A^ pathways (promoter methylation; gene mutation; LOH) are near ubiquitous events in the pathogenesis of HNSCC [59]. We have previously discussed their role in the malignant transformation of OPMD [11]. Significantly, these genetic abnormalities lead to the loss of the G1/S checkpoint and premature entry into S phase. Keratinocytes with only these alterations have minor chromosomal alterations [60], but with telomerase activation (see later), replicative senescence is bypassed [61] and chromosome fusions and dicentric chromosomes emerge [62]. Breakage of the resulting anaphase bridges leads to large amounts of cell death, but in cells that survive, there results extensive SCNA due to unbalanced chromosomal alterations and chromosomal non-disjunctions; the features are those of chromosome instability. Telomerase also inhibits DSB repair and changes cell metabolism, ROS levels, and the tumor microenvironment, amongst others, to facilitate tumor development [63]. 

Interestingly, mutant *TP53* appears to be more important than inactive *CDKN2A*/p16^INK4A^ in inducing genetic instability [64]. Furthermore, current thinking suggests that mutant *TP53* may play different roles at different stages of carcinogenesis. For example, heterozygous *TP53* mutations accumulate in ageing keratinocytes, and although the affected tissues remain histologically normal, the mutant p53 cell clones have a proliferative advantage by virtue of a bias in cell fate that favors progenitor rather than differentiated cells [65]. Clonal competition restricts the expansion of the p53 mutant population, but nevertheless, clones emerge with extensive SCNA that progress to form tumors when the second p53 allele in p53 mutant cells is lost spontaneously [65].

### 6.2. Repair

Genetic abnormalities associated with the repair of DSBs in HNSCC have been reviewed recently [57]. Whilst abnormalities in NHEJ and HR/FA genes have been described in both OSCC [47,66,67,68,69,70] and HNSCC [59,71,72], the overall picture remains unclear. 

There is a paucity of information on the status of DSB repair genes in OPMD. To address this anomaly, Farah and colleagues [73] carried out a comprehensive study to investigate gene abnormalities in the transition of OPMD to OSCC, studies that involved whole exome sequencing, enrichment analysis to characterize the effect of mutations on biological pathways, and protein expression experiments. They showed that defects in DNA repair pathways, specifically decreased expression of *BRCA1*/*BRCA2* and MMR (*MLH1*, *PMS2*, *MSH2*, *MSH6*) genes, are associated with malignant transformation in OPMD. These findings are supported by Ho et al. [74], who demonstrated that decreased expression of other components of the HR pathway (*FANCD2*, *FANCG*) were associated with malignant transformations in oral epithelial dysplasia; decreases in the phosphorylation of pFANCD2, pFANCG, pATR and pCHK-1 were also documented [74]. The transition of oral leukoplakia to OSCC in tissues of Indian origin has also been shown to be associated with defects in nucleotide excision repair, base excision repair, and mismatch repair [75]. 

### 6.3. Single Nucleotide Polymorphisms in DSB Repair Genes

Many workers have documented single nucleotide polymorphisms (SNPs) in DSB repair genes in OSCC. SNPs have been reported in both NHEJ genes [76,77,78,79,80] and HR genes [45,81,82,83,84]. SNPs are thought to account for an increased risk of both OSCC and OPMD [81,85,86,87], and whilst they may impact “disease susceptibility”, there is little, if any, information about such gene anomalies being associated with malignant transformation in OPMD. This is particularly important, because only a small minority of OPMD progress to OSCC, and outcome data are not included in the study designs. 

Studies of SNPs can be over-interpreted [88]. Often, studies are based on a small sample size, have a restricted population ethnicity, and are influenced by demographic factors (age, gender, risk factor exposure). Furthermore, what is critical is whether the SNPs cause a change in the amino acid sequence, which could result in alterations to the protein product and lead to a selective advantage for the putative cancer cells. In addition, whilst an SNP may not alter the protein sequence, it can influence the expression of a gene through its location. Many SNPs, for example, are located in the 3′ untranslated region (3′UTR) where mRNA binding motifs reside, and polymorphisms at these sites are likely to be deleterious. Some studies—but not all—meet these criteria, but functional investigations are still lacking.

It is also important to recognize that OSCC risk factors alone can influence the DDR and the occurrence of SNPs. Cigarette smoke, for example, induces phosphorylation of ATM, CHK2, ϒ-H2A.X, and p53 [89], and high and low tobacco use is known to be reflected in the occurrence of certain SNPs [90]. Ethanol also induces a DNA damage response [91]. The other primary risk factor for oropharyngeal tumors is HPV infection; the viral proteins E1/E2 activate ATM and ATR [92], and E6 and E7 proteins inhibit the cell cycle checkpoints p53 and pRB, respectively.

## 7. Conditions That Predispose to OSCC

### 7.1. Germline Mutations of HR/FA Pathway [93]

Fanconi anaemia is the most prevalent form of inherited bone marrow failure and is caused by germ-line pathogenic variants of genes in the FA pathway that repair defects in DNA inter-strand crosslinks. Patients with FA have an increased risk (700 fold) of developing HNSCC, and when it occurs, it is at a much younger age than individuals in the general population [94,95,96]. Similarly, Bloom syndrome is associated with germ-line defects in the HR pathway (mutation of the *BLM* gene), and individuals are predisposed to develop HNSCC (some 18% of secondary tumors are HNSCC) [97]. In ataxia telangiectasia (germline mutations of the ATM gene), whilst secondary leukemia and lymphoma are common, OSCC has also been reported [98]. The importance of the ATM gene in OSCC pathogenenesis is also emphasized by the fact that an ATM polymorphism (rs189037) increases the risk of OSCC [99]. Loss of heterozygosity at the ATM locus (11q23) is common in sporadic OSCC [100], and promoter hyper-methylation of the ATM gene occurs in some 25% of OSCC cases [101]. However, promoter methylation does not always equate to reduced expression (see above).

We can find no evidence of an increased incidence of OSCC in Nijmegen syndrome (mutation of NBS) or Rothman-Thompson syndrome (mutation of *RECQL4*), although in the latter condition, carcinoma of the skin is more common. 

### 7.2. Germline Mutations of NHEJ Pathway

We can find no evidence of an increased incidence of OSCC associated with germline mutations in the NHEJ pathway [93], including Werner syndrome (mutation of *WRN*), *LIGIV* deficiency, and *XLF*/*NHEJ1* deficiency. Similarly, Seckel syndrome (mutation of *ATR*) is not predisposed to forming OSCC. Furthermore, whilst it was initially thought that Li Fraumeni syndrome (mutation *TP53*) might predispose to OSCC, there has been no robust evidence in the past 20 years to support this contention [102].

### 7.3. Defects in Telomere Maintenance

Telomeres are repetitive DNA sequences at the ends of chromosomes and are sites where DNA damage is thought to persist compared to non-telomeric regions. Their role with respect to cellular senescence, ageing, and DNA damage has been reviewed recently by Eppard et al. [24], and a detailed analysis is beyond the scope of the present text. In actively dividing cells, chromosomal attrition is a natural consequence of extensive cell replication, and possibly oxidative DNA damage. Chromosomal attrition is limited by telomeres acting in conjunction with six further capping proteins collectively termed the shelterin complex. When telomeres reach a critical length, they are unable to bind sufficient capping proteins and are sensed as exposed DNA ends that results in the formation of DSBs. This phenomenon triggers a DDR that leads to cell cycle arrest and cellular senescence through the activation of downstream effector proteins p53-p21^WAF1/Cip1^ and p16^INK4A^-pRB. In non-proliferating cells, quiescent cells, or terminally differentiated cells, a DDR can be activated independently of telomere length. Telomere dysfunction combined with the activation of the DDR is thought to be a critical early event in the development of human cancer [103], a proposal that is supported by data from pre-cancer models [104], including OPMD [105,106].

The situation changes when cell cycle checkpoint pathways are inactivated (p53 and/or p16^INK4A^-Rb). In these circumstances, the uncapped telomeres are fused by NHEJ or A-EJ, leading to chromosome end-to-end fusions. Or the uncapped telomeres are processed by the HR machinery resulting in telomere alterations [107]. The cells now undergo crisis and eventually bypass cellular senescence to become immortal. The emergence of cancer cells from crisis is a key and rate-limiting step in tumor progression because the combination of short telomeres and *TP53* haplo-insufficiency results in widespread SCNA and carcinoma development [108]. Heterozygous *TP53* mutations, shorter telomeres, and telomerase deregulation are observed in keratinocytes derived from OPMD biopsies ([109]; E.K.Parkinson, J.Fleming and P.R.Harrison—unpublished data). 

To facilitate telomere homeostasis, telomere length is maintained by the activation of telomerase, a ribonucleoprotein complex that contains a catalytic telomerase reverse transcriptase subunit (*TERT*) and an integral telomerase RNA component (*TERC*; *TR* or telomerase RNA). Telomerase is activated by non-coding *TERT* promoter mutations [110,111], which have been identified in HPV-negative OSCC [112]. However, such mutations are not enough to account for the increase in telomerase in 90% of HNSCC [113]. Further genetic alterations are thought to be required, though have not been identified to date [114]. By maintaining telomere length, telomerase circumvents the development of cellular senescence and maintains genomic integrity [115]. The process is facilitated by certain DNA repair genes that bind *TERT*, *TERC*, and components of the shelterin complex to stabilize telomerase to telomeric DNA [116]. The activation of telomerase is therefore essential for the maintenance of telomere length, the avoidance of DSB formation, and the development of genomic instability; however, it is also essential for tumorigenesis [117], an observation that has been explained, albeit in part, by the extra-canonical functions of telomerase [118]. 

Telomeres are also maintained by the recombination-based alternative lengthening of telomeres (ALT) pathway that is regulated by HR/FA proteins [119,120]. ALT has been detected in normal mammalian somatic cells [121], but its function in epidermal tumorigenesis appears to be more nuanced. In *Terc* (−/−) mice, normal and SCC-derived basal keratinocytes and stem cells are ALT positive, whereas in *Terc* (+/+) mice, ALT is suppressed in primary SCCs but not in metastatic carcinomas [122]. Therefore, in the absence of telomerase, ALT is active. But when telomerase is present, ALT is only active in metastatic cells. These findings are consistent with the fact that basal keratinocytes in both mice and humans have constitutively competent ALT pathways, but there is little/no evidence of ALT activation in both OPMD and primary OSCC [59,123]. 

Dyskeratosis congenita (DC) is a disorder of excessive telomere attrition due to defects in the telomerase complex, either related to the disease gene (*DKC1*), and/or other genes involved in telomere repair (*TIN2*, *TERC*, *TERT*, amongst others [124]). The condition is characterized by accelerated ageing, nail dysplasia, abnormal skin pigmentation, oral leukoplakia, and bone marrow failure. Numerous other features have been described that reflect specific patterns of inheritance (x-linked recessive, autosomal dominance, autosomal recessive), but the most common non-hematological malignancy in DC is HNSCC, of which OSCC is pre-eminent (some 40% of secondary tumors). 

DC and FA share a number of characteristics, including bone marrow failure, short telomeres, chromosome instability, and a predisposition to form secondary solid tumors. Tummula et al. [125] reported that in conditions where there is an inherited bone marrow failure, genome instability occurs as a consequence of a primary transcription deficiency rather than a DNA repair deficit. Interestingly, there appears to be overlapping gene expression pathways between FA and DC that are particularly related to protein translation and elongation, RNA metabolism, and mitochondrial function, and which result in a common signature of 26 up-regulated genes [126]. 

It remains to be determined whether the common FA/DC genes/pathways [126] are involved in secondary OSCC formation, and whether they also play a role in the malignant transformation of OPMD. It is also unclear whether a failure of tumor-immune surveillance due to bone marrow aplasia leads to the formation of secondary solid tumors, even though the incidence of OSCC is extremely low in aplastic anaemia, an acquired form of marrow aplasia where patients are immune-compromised and cells have short telomeres [127,128].

## 8. OSCC in Young Individuals

The pathogenesis of OSCC in young patients has been extensively reviewed in the past [129,130] and is beyond the scope of the present report. Current thinking is that there is little to distinguish the genetic profile of young (<49 years; little exposure to traditional OSCC risk factors) and older (>50 years; commonly exposed to tobacco and alcohol) OSCC patients. However, it is recognized that the data are inconsistent between studies, the investigations invariably lack control groups, sample sizes are small, and there are variations in laboratory methodology. 

Cury et al. [131] recently examined DNA repair pathway genes from 45 independent and 55 TCGA-derived OSCC young patients, and demonstrated that some two thirds of these individuals had at least one germline variant in DNA repair pathway genes (primarily *ATM*, *RAD51D*, *BRCA1*, *BRCA2*), FA genes (*FANCA*, *FANCG*, *FANCM*), and DC genes (*ACD*, *TPP1*, *RTEL1*, *TERT*). These findings support previous observations concerning the FA pathway [132,133].

## 9. HR/FA Genes Are Upregulated in OPMD-Derived Keratinocyte Cultures That Have Bypassed Crisis

We have developed a broad range of keratinocyte cultures from OPMD that are either immortal or mortal in vitro [109,134,135,136,137]. Whilst both phenotypes share a number of neoplastic characteristics [134], immortal keratinocytes are resistant to suspension-induced terminal differentiation and have distinct transcriptional [137] and metabolic profiles [138]. Furthermore, certain immortal cell lines were derived from OPMD tissues that progressed to OSCC, a feature that was never present in OPMDs from which the mortal cells were derived. Importantly, immortal keratinocytes are characterized by the inactivation of *TP53* and *CDKN2A*, the activation of telomerase, and extensive SCNA and LOH. By contrast, mortal keratinocytes have wild type *TP53* and *CDKN2A*, have no gene copy number variations or gene methylation, and have few classical driver mutations [123]. There is some evidence that mortal keratinocytes evolve into their immortal counterparts [60,139,140], but recent data suggest that in some instances, they may play a more supportive role. For example, they up-regulate prostaglandin E1 and E2, which stimulate immortal keratinocyte proliferation in vitro [58]. They also over-express a variety of SASP factor transcripts and proteins [58], which have been shown extensively to mediate cancer progression [33]. Current thinking is that the mortal OPMD cells are normal damaged keratinocytes that are approaching senescence [58,123,138]. 

We mined a previously published Affymetrix gene expression database to examine the expression of specific DSB repair genes in the above OPMD-derived mortal and immortal cells, as well as in normal oral keratinocytes [137]. *BLM*, *BRCA1*, *RAD51C*, and *PRKDC*, together with other DNA repair genes, were significantly over-expressed in the immortal keratinocytes relative to the OPMD-derived mortal keratinocytes and normal oral keratinocytes. There was no evidence of reduced levels of expression in any of the above DNA repair genes in the mortal OPMD keratinocytes and normal keratinocytes. Although these changes were less than two-fold and therefore not reported in our original microarray study [137], they were highly significant. Interestingly, the genes that were most convincingly over-expressed in the immortal OPMD keratinocytes were involved in the HR/FA pathway (*BLM*, *BRCA1*/*FANC*, BRAC2/*FANCD1*, *RAD51C*/*FANCR*, *XRCC2*/*FANCU*) (Figure 1).

To test whether the up-regulation of DNA repair was due to the faster cycling rate of immortal OPMD keratinocytes, we examined the cell cycle status by mining data on the cyclins and their associated Cdks. Mortal OPMD keratinocytes had much higher ratios of cyclin D1/cyclin A, cyclin D1/cyclin E1, and cylinD1/Cdk1 relative to the normal and immortal OPMD keratinocytes, findings that are consistent with a reduced passage through the restriction point and entry into S phase [58,138]. We then conducted linear regression analysis to determine whether the elevated levels of the DNA repair genes correlated with specific phases of the cell cycle. There was a strong correlation between most DNA repair genes and cyclinB1/Cdk1 (Figure 2); the correlation was much less with cyclin E1 and even less with cyclin A1 and cyclin D1. The data suggest that the elevation of DNA repair gene transcription occurred primarily during S and G2.

We examined the expression of DNA repair genes in D17, an oral keratinocyte cell line with wild type p53, absence of p16^INK4A^, and minimal chromosome alterations. Whilst the data needs verification in a much larger dataset, we found that D17 did not have higher levels of any of the DNA repair genes relative to normal oral keratinocytes. However, when we examined the transcript levels of the same DNA repair genes in D17 cells immortalized by the ectopic expression of telomerase, *BRCA1* and *PRKDC* were higher than normal, suggesting that the up-regulation of these two DNA repair genes may be associated with the bypass of telomere dysfunction and telomerase deregulation (Figure 3).

## 10. Discussion

In the present study, we examined whether a failure to repair DNA DSBs contributed to the development of OSCC. We conclude that the HR/FA DNA repair pathway that is involved in the repair of DSBs and interstrand crosslinks is pivotal in the development of OSCC. First, defects in HR/FA genes are associated with the malignant transformation of OPMD. Second, germline mutations in the HR/FA pathway, but not NHEJ genes, predispose to OSCC. DC also predisposes to OSCC and is caused by inherited defects in the telomerase complex. Third, abnormalities in HR/FA genes are prevalent in young patients with OSCC who often have minimal exposure to traditional OSCC risk factors. Fourth, we present new data showing overexpression of HR/FA genes in OPMD-derived immortal keratinocytes compared to their mortal counterparts. Certain immortal keratinocyte cell lines were derived from OPMD tissues that progressed to OSCC, whereas in our most recent studies, this was never a characteristic of OPMD from which mortal keratinocytes were derived [109,137].

The overexpression of HR/FA genes in this study (*BLM*, *BRCA1*, *BRCA2*, *RAD51C*) occurred primarily in the S and G2 phases of the cell cycle, confirming that HR gene expression is largely mediated by cell cycle regulation. Interestingly, this is a period when the classical transactivation-dependent function of p53 is operative [141], and previous work has shown that whilst wild type p53 suppresses HR [142], mutant p53 stimulates HR expression [143]. However, the S/G2 phase of the cell cycle is also the time of telomerase activation [144]. Our results with D17 cells, are therefore particularly interesting, because the ectopic expression of telomerase in D17 (wild type p53, deletion p16^INK4A^) resulted in the up-regulation of *BLM*, *BRCA1*, *BRCA2*, and *RAD51C*. The findings suggest that HR/FA gene overexpression may be a response to telomere dysfunction and telomerase deregulation. The up-regulation of DNA repair genes in cancer is not uncommon [145] but, as far as we are aware, the up-regulation of HR/FA genes has not been reported previously relative to the stage of tumor progression or progression to replicative immortality.

In the present study, we report HR/FA gene overexpression in OPMD-derived immortal keratinocytes suggesting enhanced DNA repair capacity, whereas in other studies involving OPMD and OSCC, gene mutation indicative of loss of gene function has been documented. These observations are not mutually exclusive, because gene mutations can affect the regulation of gene expression whether in terms of gene transcription, mRNA stability, or gene translation [146,147]. We believe that gene overexpression in the present study is occurring in response to increasing levels of SCNA and genomic instability, but further work is required to determine whether gene overexpression occurs before or after gene mutation. What is clear, however, is that an alteration of DNA repair capacity is fundamental in the malignant transformation of OPMD. 

Both gene mutations and copy number alterations (gains and losses) have been described in a wide variety of sporadic cancers of different origin (65% of 10,202 cancers from the TCGA database; [148]), with overexpression being attributed to coordinated regulation through the Rb/E2F pathway [149]. Therefore, it may be that FA genes fulfill two roles in cancer pathogenesis. On the one hand, gene mutation would likely lead to genomic instability and the possible development of a mutator phenotype, the result being an increase in cellular diversity followed by clonal selection. On the other hand, elevated gene expression would accommodate further DNA damage due to increased replication stress, and thus would act as a survival factor. Taken together, we believe anomalies in the HR/FA pathway provide a selective advantage to putative cancer cells (Figure 4). 

In this study, we examined the expression of *PRKDC* and *LIGIV* of the NHEJ pathway and found increased and decreased expression, respectively, in the immortal OPMD-derived keratinocytes. The significance of these findings is unclear. Sishc and Davis [33] comprehensively reviewed the role of NHEJ in carcinogenesis and concluded that defects in NHEJ genes in rodent models of carcinogenesis resulted in genomic instability and the promotion of carcinogenesis, but in humans, mutations of NHEJ genes were rare and down-regulation of NHEJ factors occurred only in a small number of cancers. Over-expression of NHEJ factors (particularly Ku70 and *PRKDC*), however, occurred commonly in a broad spectrum of cancers [42]. 

Is the HR/FA pathway essential in the pathogenesis of sporadic OSCC? Three of the most common risk factors in sporadic OSCC are tobacco, alcohol, and areca alkaloids, all of which are known to be DNA-damaging agents [150,151,152]. The HR/FA pathway is therefore likely to play a protective role in combating these DNA insults. It follows that defects in the HR/FA pathway will be pivotal in the aetiology of OSCC, and to support this line of thinking, downregulation of FA gene expression, epigenetic silencing of FA genes, copy number alterations, and somatic mutations of FA genes have been reported in HNSCC [59,69,70]. Furthermore, Verhagen et al. [153] reported bi-allelic germline and somatic HR/FA variants in 19% of sporadic HNSCC and established that such variants resulted in a functional deficit in the HR/FA pathway. It is also interesting to note that HR/FA signalling intersects with the glucocorticoid receptor (GR) pathway, which is critically involved in carcinogenesis through what is known as the cancer-associated glucocorticoid system [154]. For example, GR finely regulates *BRCA1* expression and impacts DNA repair capacity [155]. Thus, the incidence of a defective HR/FA pathway appears to be relatively common in HNSCC, and therefore warrants further consideration. This may have important consequences, because with the advent of high-throughput gene sequencing, the inclusion of HR/FA gene pathways with driver gene molecular profiles may facilitate a more accurate prediction of the malignant transformation of oral cancer-precursor lesions.

## Figures and Tables

**Figure 1 ijms-25-04092-f001:**
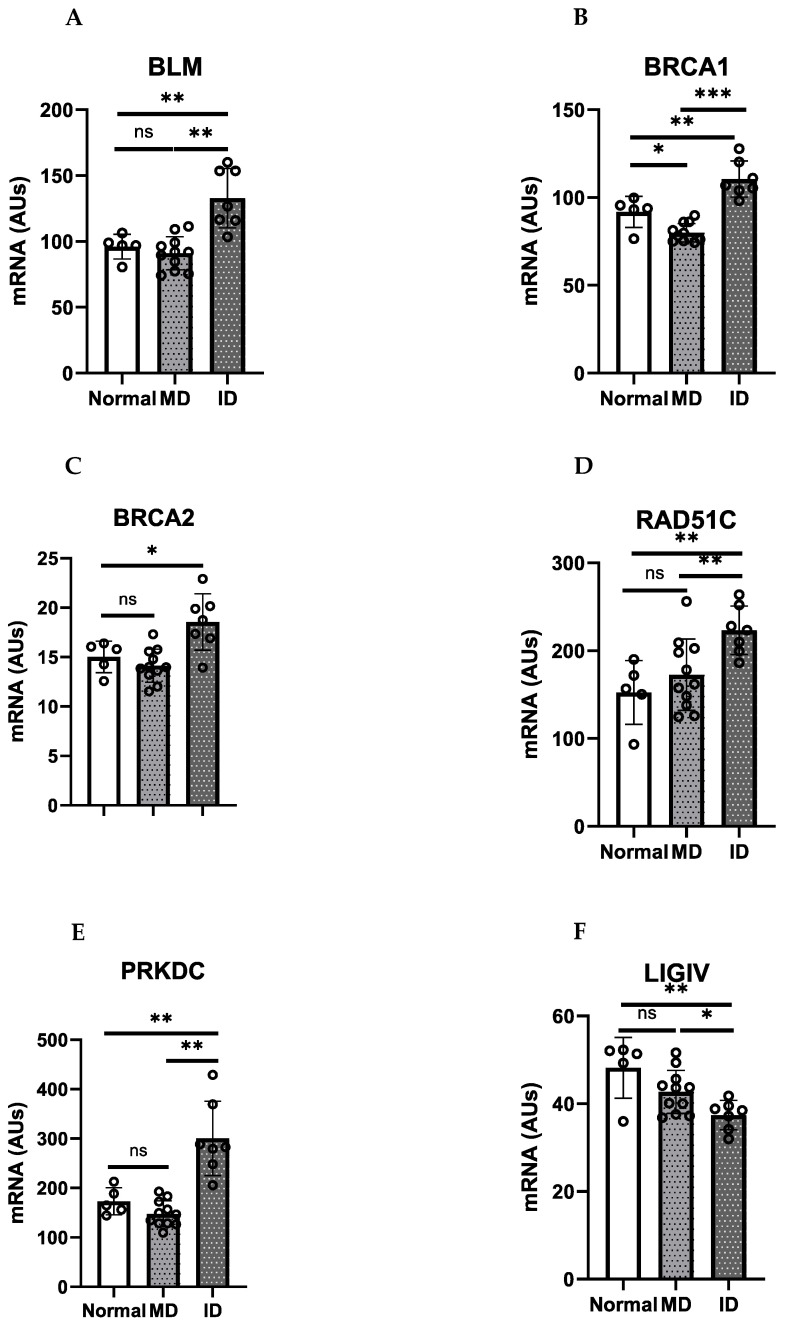
DNA HR repair genes associated with early onset oral cancer are upregulated in immortal dysplasia keratinocytes. (**A**). *BLM*; (**B**). *BRCA1*; (**C**). *BRCA2*; (**D**). *RAD51C*; (**E**). *PRKDC*; (**F**). *LIGIV*. Normal (*n* = 5), mortal dysplasia (MD; *n* = 11), immortal (ID; *n* = 7). Data are means +/− standard deviation. Data derived from one published microarray experiment [128]. ns, not significant; *, *p* < 0.05; **, *p* < 0.01; ***, *p* < 0.005.

**Figure 2 ijms-25-04092-f002:**
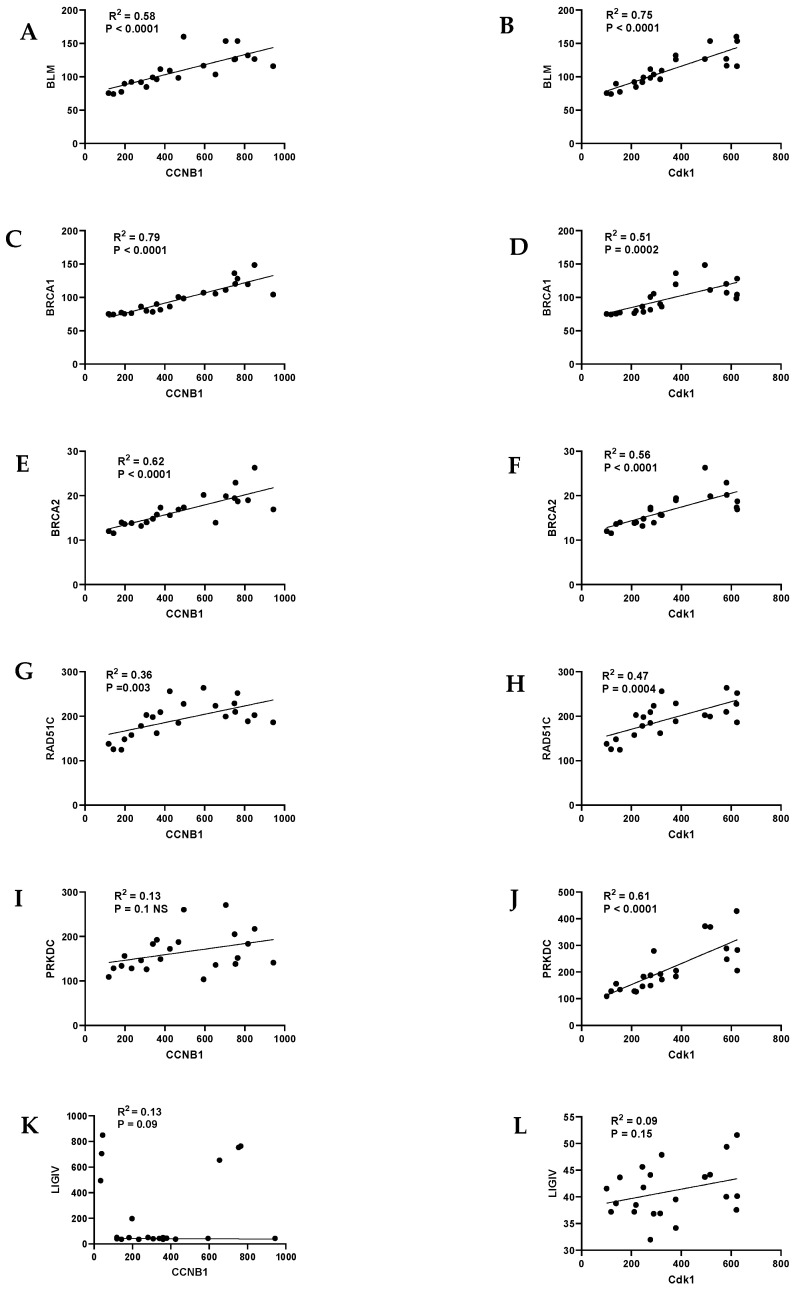
DNA HR repair genes associated with early onset oral cancer correlate with S and G2/M cyclin/cdk complex transcripts cyclin B1/cdk1 in dysplasia keratinocytes. The left panel shows the linear regression analysis of the DNA HR repair gene transcripts versus cyclin B1 (*CCNB1*) and the right hand panel shows a similar analysis of the same gene transcripts versus Cdk1. (**A**,**B**), *BLM*; (**C**,**D**), *BRCA1*; (**E**,**F**), *BRCA2*; (**G**,**H**), *RAD51C*; (**I**,**J**), *PRKDC*; (**K**,**L**), *LIGIV*. R2 and *p* values are indicated in each figure. Data derived from one published microarray experiment [128].

**Figure 3 ijms-25-04092-f003:**
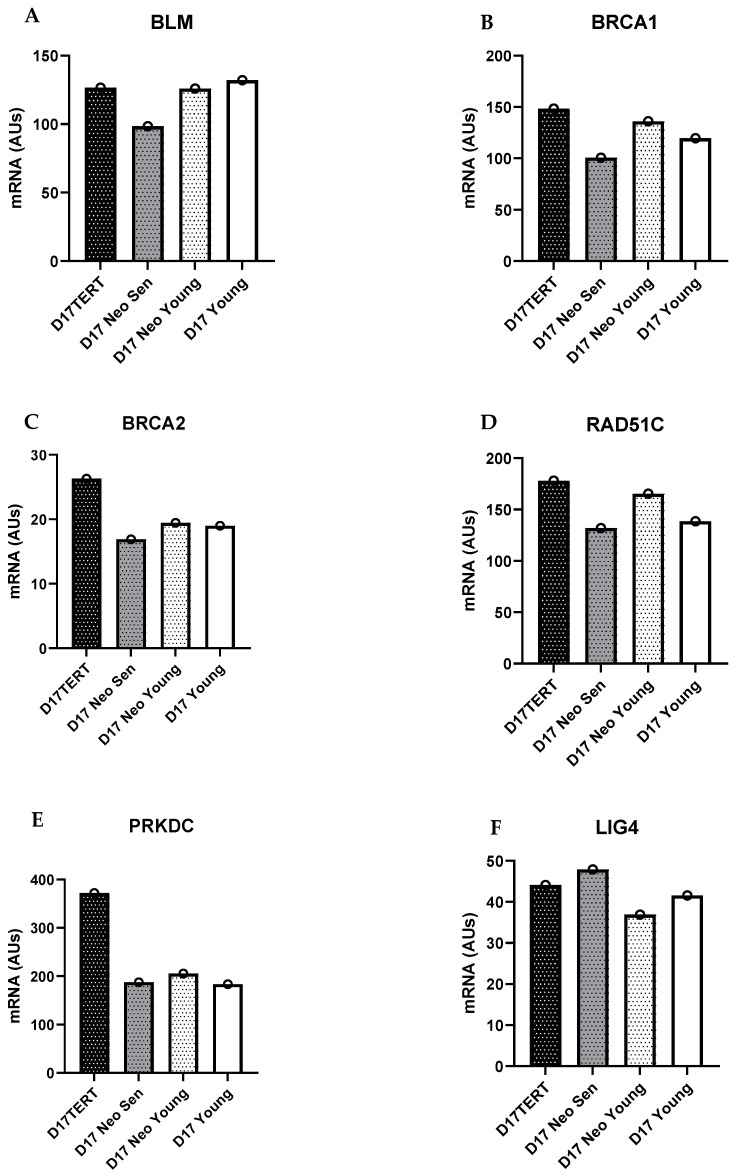
NHEJ repair gene LIG4 is downregulated in immortal dysplasia keratinocytes. (**A**). Young control (white bars), young (neo vector control; white stippled bars) senescent (neo vector contro; dark stippled bars), and *TERT* neo-immortalised D17 cells (dark bars. (**A**). *BLM*; (**B**). *BRCA1*; (**C**). *BRCA2*; (**D**). *RAD51C*; (**E**). *PRKDC*; (**F**). *LIGIV*. Data are means +/− standard deviation. Data derived from one published microarray experiment [128]. (*n* =1 because only one line similar to D17 was available).

**Figure 4 ijms-25-04092-f004:**
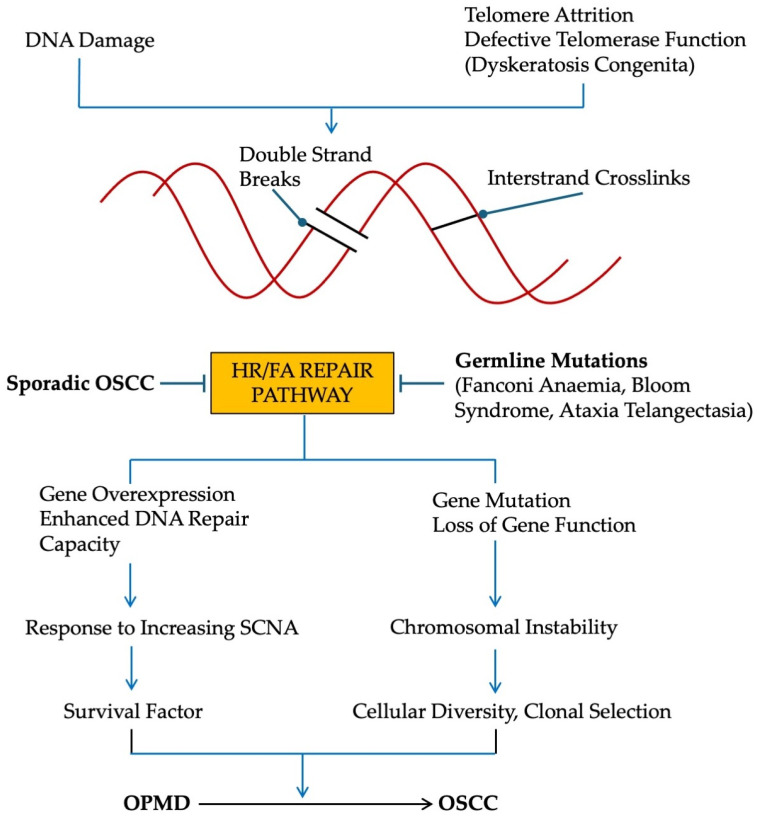
Schematic of the possible effects of a defective HR/FA pathway during the development of oral cancer. (HR, Homologous recombination; FA, Fanconi anaemia; OPMD, oral potentially malignant disorder; OSCC, oral squamous cell carcinoma; SCNA, somatic copy number alterations.

**Table 1 ijms-25-04092-t001:** Genes and proteins involved in the repair of DSBs.

Homologous Recombination (HR)	Non-Homologous End Joining (NHEJ)
*RAD51*: encodes RAD51	*XRCC5*: encodes Ku80 *XRCC6*: encodes KU70
*XRCC2:* encodes XRCC2 *XRCC3*: encodes XRCC3	*PRKDC and XRCC7:* encode DNA-dependent protein kinase (PK) catalytic subunits
*TP53:* encodes p53	*DCLRE 1C:* encodes Artemis protein which acts as an endonuclease
*RPA1:* encodes Replication Protein A (RPA)	*POLM:* encodes DNA polymerase Pol μ *POLM:* encodes DNA polymerase Pol λ
*BRCA1* *BRCA2*	*XRCC4/LIGIV*: encodes Ligase IV
*BLM:* encodes DNA helicase RecQ	
*MUS81*: encodes endonuclease enzyme	

## Data Availability

Data are available upon reasonable request to the corresponding authors. Data mining was undertaken on previously published data.

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
