# Peer review of "A Review of the Repair of DNA Double Strand Breaks in the Development of Oral Cancer"

_ijms, 2024, doi:10.3390/ijms25074092_

Round 1
Reviewer 1 Report
Comments and Suggestions for Authors
This is a nice summary of the repair of DNA double strand breaks in the oral cancer development with extensive literature review. The manuscript is well written and nicely organized. My only question is: Can authors add statistical analysis for figure 3 as they did for figure 1?
Author Response
Referee 1
We are grateful for the complimentary comments of this Referee.
Figure 3: It is not possible to provide statistical analysis because only one experiment was undertaken, as indicated in the legend. If this proves to be a problem, we are happy to remove the figure and rely on the text.
Reviewer 2 Report
Comments and Suggestions for Authors
The authors have prepared an interesting publication, but the question arises: where do the authors see the role of ROS in the formation of DSBs and genetic instability, as well as cancer transformation and the development of oral cancer? Oral cancer is accompanied by increased ROS levels. Cancer cells are generally characterized by DDR and increased levels of ROS. Genetic changes lead to an increase in the production of ROS, which is an endogenous source of DSBs.
The abbreviation DDR was not explained when it first appeared in the text.
There are too many self-quotes in the publication.
Author Response
please find attached our response letter

Reviewer 3 Report
Comments and Suggestions for Authors
Prime and colleagues presented a comprehensive review article on the role of double-strand breaks in inducing the neoplastic transformation of oral lesions. The authors well-described the pathogenetic mechanisms mediated by DSBs as well as the pathways and mechanisms involved in the development of neoplastic lesions. Although interesting, the manuscript needs some revisions before publication:
1) When describing the epidemiology of OSCC, please add also references specific to oral lesions (add refs to Ref 1). For this purpose, please see:
- PMID: 37676656
- PMID: 37064317
- PMID: 36005828
2) Before describing the consequences of DSBs (Chapter 3), you have to introduce the causes responsible for DSBs, including tobacco smoking, alcohol, oral microbiota dysbiosis, chronic inflammation, etc.;
3) In some parts, the review is verbose and redundant. The length of the manuscript should be reduced. Please consider to merge chapter 4 and chapter 5 and reduce the content;
4) In the Discussion section, please briefly mention how the novel high-throughput may be used for the identification of pre-cancerous lesions and molecular alterations and predict the development of OSCC.
Author Response

(The authors gave the same response as above.)

Round 2
Reviewer 3 Report
Comments and Suggestions for Authors
Dear Authors,
all of my previous comments were properly addressed. The manuscript can be accepted for publication after the editorial check.